# GntR-like SCO3932 Protein Provides a Link between Actinomycete Integrative and Conjugative Elements and Secondary Metabolism

**DOI:** 10.3390/ijms222111867

**Published:** 2021-11-01

**Authors:** Krzysztof J. Pawlik, Mateusz Zelkowski, Mateusz Biernacki, Katarzyna Litwinska, Pawel Jaworski, Magdalena Kotowska

**Affiliations:** Hirszfeld Institute of Immunology and Experimental Therapy, Polish Academy of Sciences, 53-114 Wroclaw, Poland; mz548@cornell.edu (M.Z.); mateusz.biernacki@selvita.pl (M.B.); katarzyna.litwinska@carpegen.de (K.L.); paweljaworski1988@gmail.com (P.J.); magdalena.kotowska@hirszfeld.pl (M.K.)

**Keywords:** streptomyces, specialized metabolism regulation, AICEs, GntR, HutC, KorSA, kil-kor system

## Abstract

*Streptomyces* bacteria produce a plethora of secondary metabolites including the majority of medically important antibiotics. The onset of secondary metabolism is correlated with morphological differentiation and controlled by a complex regulatory network involving numerous regulatory proteins. Control over these pathways at the molecular level has a medical and industrial importance. Here we describe a GntR-like DNA binding transcription factor SCO3932, encoded within an actinomycete integrative and conjugative element, which is involved in the secondary metabolite biosynthesis regulation. Affinity chromatography, electrophoresis mobility shift assay, footprinting and chromatin immunoprecipitation experiments revealed, both in vitro and in vivo, SCO3932 binding capability to its own promoter region shared with the neighboring gene *SCO3933*, as well as promoters of polyketide metabolite genes, such as *cpkD*, a coelimycin biosynthetic gene, and *actII-orf4*—an activator of actinorhodin biosynthesis. Increased activity of SCO3932 target promoters, as a result of SCO3932 overproduction, indicates an activatory role of this protein in *Streptomyces coelicolor* A3(2) metabolite synthesis pathways.

## 1. Introduction

Streptomycetes are morphologically complex Gram positive soil bacteria that produce a variety of secondary metabolites. Many of these compounds have antibiotic, antifungal, anticancer, immunosuppressant and other biological activities [1]. As revealed by genomic data, virtually each *Streptomyces* strain has the potential to produce between several and several dozens of secondary metabolites. However, only a fraction of them can be detected in the laboratory conditions due to strict and complex regulatory mechanisms governing the expression of biosynthetic gene clusters (BGCs). The production of secondary metabolites is controlled at multiple levels and is closely related to other cellular events, such as mycelial growth, hyphae development, sporulation and others [2,3,4]. 

Genome of *Streptomyces coelicolor* A3(2), the model strain in the studies on the genus *Streptomyces*, encodes over 20 BGCs [5]. Genetic studies of this organism are facilitated by its production of colored secondary metabolites: red undecylprodigiosin (RED), blue actinorhodin (ACT) [1] and, the last to be discovered, yellow coelimycin (CPK) [6,7]. 

The biological roles of the yellow compounds identified as coelimycins P1 and P2 [8] are not fully understood. They are formed in the culture medium by a non-enzymatic reaction of *N*-acetylcysteine and glutamate, respectively, with the predicted *bis*-epoxide coelimycin A [9], deduced to be the weak antibiotic abCPK reported by Gottelt et al., 2010 [7]. Its polyketide backbone is synthesized by a type I modular polyketide synthase Cpk. 

Regulation of *cpk* gene cluster expression was recently reviewed in detail by our group [10]. It is switched on under specific culture conditions (rich medium without glucose [6] or minimal medium supplemented with glutamate) for a short time at the transition phase of growth, around 20–24 h of growth [11]. Two cluster situated activators from the family of *Streptomyces* antibiotic regulatory proteins (SARPs), CpkO and CpkN, are required for CPK biosynthesis [7,12]. They are controlled by the butanolide system proteins, which are also encoded within the *cpk* gene cluster [12,13,14]. Expression of *cpk* gene clusters is also influenced by several pleiotropic regulators which respond to environmental signals and coordinate secondary metabolism with the nutritional state of the colony [10]. Moreover, cluster situated regulators (CSRs) dedicated mainly to control their own gene clusters can also influence expression of other BGCs [12,15].

The study reported here was intended to identify unknown regulators of transcription of the *cpk* gene cluster. Surprisingly, we found SCO3932, a GntR-like protein encoded within a pSAM2-like insertion sequence in the chromosome of *S. coelicolor* A3(2), to be involved in the control of both CPK and ACT production. SCO3932 is a homologue of KorSA associated with the maintenance of the pSAM2 integrative and conjugative element from *Streptomyces. ambofaciens* [16].

Integrative and conjugative elements (ICEs) of bacterial genomes have both plasmid- and bacteriophage-like features. They have the ability to excise, conjugate to a new host and integrate in the host chromosome by site-specific recombination. Actinomycete integrative and conjugative elements (AICEs) additionally have the ability to replicate autonomously like plasmids.

This work presents a regulatory link between AICEs and the secondary metabolism of *Streptomyces*.

## 2. Results

### 2.1. SCO3932 Is a Potential Regulator of Cpk Cluster

Cpk synthase gene cluster covering genes SCO6269-SCO6288 [17] contains at least eight transcriptional units. A number of regulatory proteins, both cluster situated and pleiotropic, take part in the regulation of coelimycin synthesis [10]. Searching for new regulatory factors we decided to screen for proteins bound to the promoter regions of genes encoding the major synthase subunit CpkA and genes participating in tailoring reactions (*cpkDEG*) and transport (*cpkF*). The DNA region located between *cpkA* and *cpkD* genes amplified with AD1-BT (biotinylated) and AD2 primers was bound to magnetic beads and used as a promoter bait fragment for potential binding of regulatory proteins. A biotinylated fragment amplified with CTR1-BT and CTR2 primers was used in parallel as a non-specific control of binding. DNA-affinity chromatography was performed with lysates of *S. coelicolor* M145 from the 19th hour of growth (the logarithmic phase before the onset of coelimycin synthesis) and the 41st hour of growth (the stationary phase). Among the proteins from the 19th hour bound to *cpkA-D* promoter and washed out with 150 mM NaCl, we have found a unique 28 kDa protein band, which was not observed in the control sample. Mass spectrometry analysis of the cut-out band revealed five proteins (Table 1). Two of them have significant scores but only SCO3932 protein has molecular mass corresponding to the gel extracted fragment. 

The gene *SCO3932* is located within a pSAM2-like insertion sequence—AICE remnant *SCO3937* [18]. SCO3932 is a homolog (amino acid similarity 44%, identity 30%) of KorSA protein from *Streptomyces ambofaciens* pSAM2 mobile element [16]. Other proteins encoded within this AICE remnant homologous to pSAM2 gene products are SCO3933 homologous to Pra regulatory protein (similarity 50%, identity 35%), SCO3934 homologous to TraSA (similarity 54%, identity 42%) and SCO3936 homologous to RepSA (similarity 56%, identity 44%). SCO3937 is a serine integrase, different from tyrosine integrases typical for AICEs [18].

The protein SCO3932 belongs to the HutC subfamily [19] of the GntR family of transcriptional regulators [20]. It contains a helix–turn–helix DNA binding domain and a ligand binding UTRA (UbiC transcription regulator-associated) domain [21].

### 2.2. Protein SCO3932 Binds to Its Own Promoter and to Promoters of Secondary Metabolism-Related Genes

SCO3932 protein with His-tag on the C-terminus was produced in *E. coli* pET expression system and used for electrophoretic mobility shift assay (EMSA) with fragments pcpkA/D, as well as p3932/3933 and pactII-orf4 (Table 2). Promoter of *actII-orf4* (actinorhodin biosynthesis activator, [22] was included in our study, because it was reported earlier by Park et al., 2009 [23] as a binding target for the SCO3932 protein. Among the tested fragments, the promoter region p3932/3933 was bound with the highest affinity by SCO3932 protein (Figure 1a) and fragments pcpkA/D and pactII-orf4 were bound with lower affinity (Figure 2a and Figure 3a). The 1.9 nM concentration of the protein was enough to form noticeable complexes with p3932/3933, while in case of *cpkD* and *actII-orf4* promoters concentration 9.4 nM was needed. Similarly, in the DNase I footprint we observed protection of SCO3932/SCO3933 promoter fragments in 100 nM protein but promoters of *cpkD* and *actII-orf4* were protected in 800 nM protein concentration.

Fragments positively selected in EMSA assays were used to establish the binding sequence in footprinting experiments (Figure 1, Figure 2 and Figure 3). In the p3932/3933 fragment a large area was protected from digestion with DNase I, so a DMS footprint was also performed. The results of the DMS footprint confirmed the results of the DNase I footprint. Overall, five protected regions were identified (sites K1 and K2 in the intergenic region between *SCO3932* and *SCO3933* genes, site D upstream of *cpkD*, sites A1 and A2 in the promoter region of *actII-orf4*. The consensus sequence motif AC[TC]T[AG]TT[GC][TG][GT]ACG was found with the MEME algorithm [24] (https://meme-suite.org/meme/tools/meme accessed on 16 June 2021). Interestingly, in the site K2 the motif is repeated, which is consistent with the strongest binding of this fragment by the protein.

**Figure 1 ijms-22-11867-f001:**
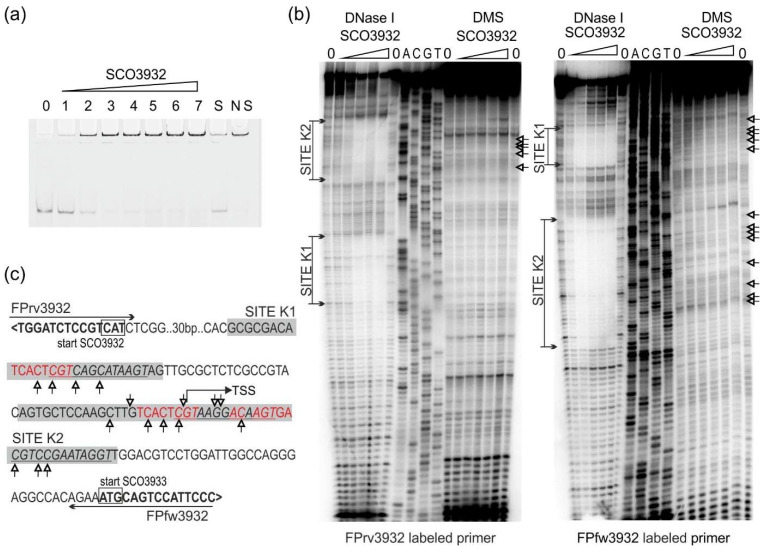
Binding of SCO3932 protein to the intergenic region between *SCO3932* and *SCO3933* genes. (**a**) Electrophoretic mobility shift assay with p3932/3933 fragment. Lanes 0–7 contain 0, 0.9, 1.8, 9.4, 18, 28, 47 and 94 nM SCO3932 protein, respectively. Lanes S and NS contain ten-fold excess of the specific and nonspecific competitor DNA, respectively, and 94 nM SCO3932 protein; (**b**) DNase I and DMS footprinting of SCO3932 binding to SCO3932 and SCO3933 promoter region. The concentrations of SCO3932 protein used were 0, 50, 100, 200, 400, 800, 0 nM in the consecutive lines. Lanes G, A, T, C correspond to the sequence ladder generated with the labeled primer. The regions protected from cleavage by DNase I are indicated (sites K1, K2). On DMS footprint guanines protected from cleavage are marked with small arrows; (**c**) DNA sequence of the SCO3932 and SCO3933 promoter region. Primers are marked with horizontal arrows. Coding sequences are in bold, and gene orientation is shown with <, >. Translational start codons are marked with boxes. Bent arrow denotes transcriptional start site (TSS) according to Jeong et al., 2016 [25]. The sequences protected by SCO3932 from cleavage (sites K1, K2) are indicated by the shaded boxes. Protected guanines are marked with small arrows. Sequence repeats (with one mismatch) are in red. Sequence corresponding to the common motif identified by MEME (see Figure 4) is underlined and in italic.

**Figure 2 ijms-22-11867-f002:**
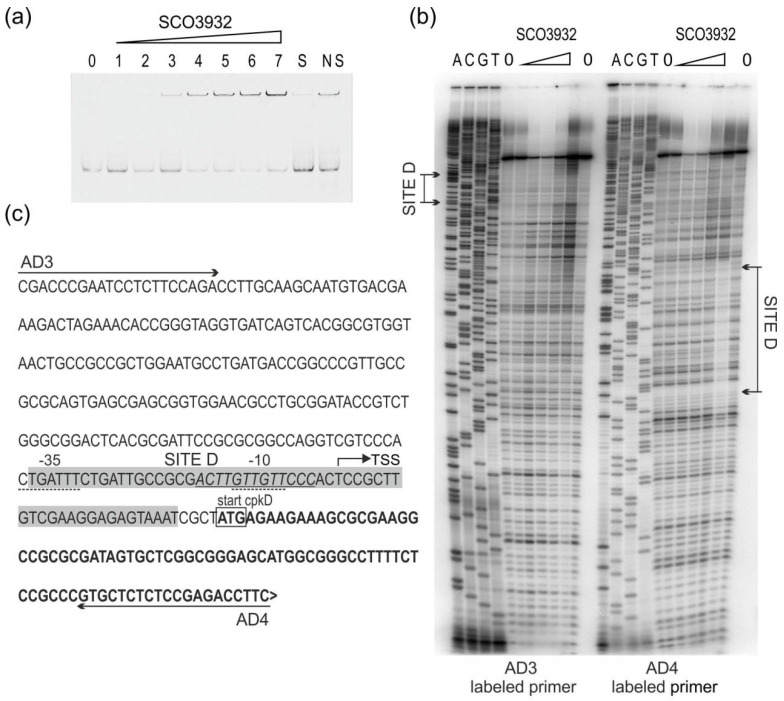
Binding of SCO3932 protein to the intergenic region between *cpkA* and *cpkD* genes. (**a**) Electrophoretic mobility shift assay with pcpkA/D fragment. Lanes 0–7 contain 0, 0.9, 1.8, 9.4, 18, 28, 47 and 94 nM SCO3932 protein, respectively. Lane S—ten-fold excess of the specific competitor DNA, lane NS ten-fold excess of nonspecific competitor DNA, respectively, and 94 nM SCO3932 protein; (**b**) DNase I footprinting of SCO3932 binding to the *pcpkD* promoter region. The amounts of SCO3932 used in the footprint were 0, 100, 200, 400, 800 1600, 0 nM. The region protected from cleavage by DNase I is indicated (site D). Lanes G, A, T, C correspond to the sequence ladder generated with the labelled primer. (**c**) DNA sequence of the *cpkD* promoter region. Primers are marked with horizontal arrows. Coding sequence is bold, translational start codon is marked with a box. Bent arrow denotes transcriptional start site (TSS) according to Jeong et al., 2016 [25]. The −35 and −10 regions are underlined with a dashed line. The sequence protected by SCO3932 from cleavage (site D) is indicated by the shaded box. Sequence corresponding to the common motif identified by MEME (see Figure 4) is underlined and in italic.

**Figure 3 ijms-22-11867-f003:**
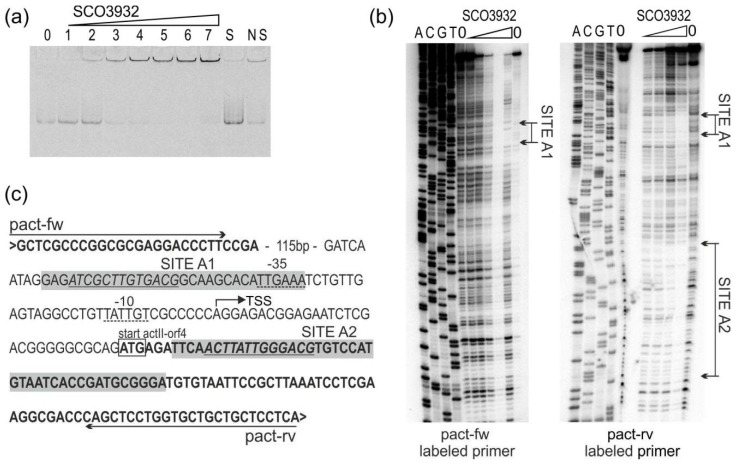
Binding of SCO3932 protein to the promoter region of *actII-orf4.* (**a**) Electrophoretic mobility shift assay. Lanes 0–7 contain 0, 0.9, 1.8, 9.4, 18, 28, 47 and 94 nM SCO3932 protein, respectively. Lanes S and NS contain ten-fold excess of the specific and nonspecific competitor DNA, respectively, and 94 nM SCO3932 protein; (**b**) DNase I footprinting of SCO3932 binding to the *actII-orf4* promoter region. The amounts of SCO3932 used were 0, 100, 200, 400, 800 1600 nM. The regions protected from cleavage by DNase I are indicated (sites A1, A2). Lanes G, A, T, C correspond to the sequence ladder generated with the labelled primer. (**c**) DNA sequence of the *actII-orf4* promoter region. Primers are marked with horizontal arrows. Coding sequence is bold, translational start codon is marked with a box. Bent arrow denotes transcriptional start site (TSS) according to Jeong et al., 2016 [25]. The −35 and −10 regions are underlined with a dashed line. The sequences protected by SCO3932 from cleavage (sites A1, A2) are indicated by the shaded boxes. Sequence corresponding to the common motif identified by MEME (see Figure 4) is underlined and in italic.

**Figure 4 ijms-22-11867-f004:**
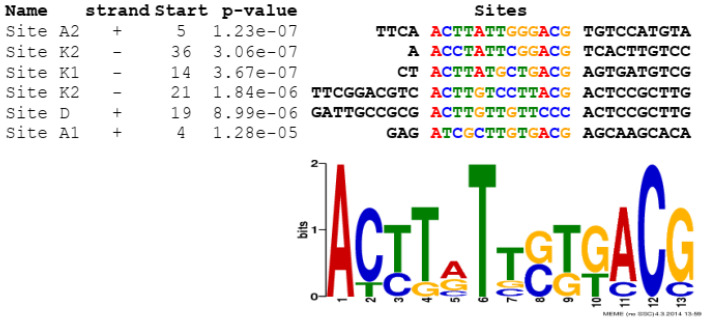
Common sequence motif found by the MEME algorithm within the SCO3932 binding sites identified in DNase I footprint experiments.

### 2.3. Binding of SCO3932 to Its Targets In Vivo

To clarify the influence of SCO3932 protein on the secondary metabolite production, two over-expression strains were constructed. Strain P170 has a multicopy plasmid pKL20 carrying SCO3932 gene under the control of ermEp* constitutive promoter [26], while P171 contains an extra copy of SCO3932 gene under the thiostrepton inducible promoter PtipA on the integrated pKL21 construct (Table 2). SCO3932 gene knock-out was also undertaken, however multiple attempts of gene deletion were unsuccessful. We tried to perform deletion by homologous recombination with the vector pMB14 containing two homology arms and a neomycin resistance gene. After conjugation we repeatedly obtained a few bald *S. coelicolor* colonies which did not survive the passage. The results suggest that deletion of SCO3932 gene may be lethal to *S. coelicolor* A3(2).

P171 strain with SCO3932 overexpression induced by thiostrepton (8 μg/mL), as well as wild-type strain M145 were used for ChIP-PCR experiment. Immunoprecipitation of formaldehyde-crosslinked complexes of protein SCO3932 bound to chromosomal DNA was conducted, followed by PCR on the obtained DNA samples, comparing it to a control genomic DNA (Figure 5). When SCO3932 was overexpressed, fragments covering promoter regions of genes SCO3932/SCO3933, *cpkD* and *actII-ORF4* could be found in the sample, while in the sample from the wild-type strain only the SCO3932/SCO3933 promoter was clearly detected by PCR. The other two were barely visible as faint bands. SCO3932 protein is capable of binding to all three investigated fragments in vivo.

### 2.4. Impact of SCO3932 Overexpression on Secondary Metabolism

The influence of SCO3932 constitutive overexpression on the activity of *actII-orf4* and *cpkD* promoters was investigated in the luciferase reporter system as described before in Bednarz et al., 2021 [12]. Both promoters were found to be more active in SCO3932 overexpression strain P170 than in the control strain P123 with an “empty” control plasmid (Figure 6a). Phenotypic observations showed that ACT production was enhanced in the strain P171, in which SCO3932 overproduction was induced, in comparison to the control strain P106. CPK secretion was similar in both strains (Figure 6b).

## 3. Discussion

### 3.1. Role of SCO3932 within the AICE

Many bacteria have in their genomes mobile elements which can be excised, transferred to another cell by conjugation and integrated into the recipient’s genome by site-specific recombination. This broad class of mobile genetic elements (MGEs) is called integrative and conjugative elements (ICEs). Actinomycete integrative and conjugative elements (AICEs) are a special type of the above-described elements which have both plasmid- and bacteriophage-like features. In addition to the basic features, they have the ability to self-replicate in the circular form before conjugation [18]. One of the best characterized AICEs is pSAM2 of *Streptomyces ambofaciens* [16,27]. 

In a search for new regulatory proteins controlling the coelimycin biosynthetic gene cluster of *S. coelicolor* A3(2) we have identified a SCO3932 protein encoded within a pSAM2-like insertion sequence—AICE remnant SCO3937 [18]. Our results indicate that this protein has several functions. It plays a role not only within the AICE, but also in the regulatory network controlling secondary metabolism.

The characteristic feature of the transfer of a certain AICEs is the creation of pocks—the cell growth inhibition areas in the plate culture of the recipient. To date, this phenomenon is unclear; it is believed that the regulatory system called kil-kor inhibits the growth of the recipient until the formation of a sufficient number of copies of the plasmid [28]. The genes assigned as *kil* have lethal effect on the host when overexpressed. The *kor* (for kil override) element includes one or more genes encoding proteins which either directly or indirectly control the expression level of the kil and thus protect the cell. 

In pSAM2, the kil-kor system is associated with transfer functions, where *traSA* and *korSA* genes are the main kil and kor elements, respectively. An effect of the kil-kor system activity in pSAM2 is a failed attempt to delete *korSA* [29]. After transformation, the mutants grew much more slowly (ten days compared with the others which typically needed two days), and after a passage to fresh medium they did not form new colonies. Similarly, in the current work we were unable to delete the *SCO3932* gene, which is a *korSA* homologue. This suggests that SCO3932 protein functions as a *kor* element of the AICE remnant SCO3937, controlling the unknown *kil* gene, possibly the *traSA* homologue—*SCO3934*. 

KorSA is a repressor [28]. It binds to its own promoter, which is a common feature of the GntR family proteins, and to the *pra* gene promoter [29]. Pra protein is a transcription activator of the three genes in the operon lying on pSAM2—*repSA*, *int* and *xis*. These genes encode a replicase, an integrase and a protein responsible for pSAM2 excision, respectively [30]. Interestingly, no binding site of KorSA was found upstream of *traSA*. Thus, KorSA seems to control the main functions of the pSAM2 genetic element via the *pra* gene [29].

In pSAM2, the *pra* gene is located downstream from *korSA*. Here, *SCO3932* (*kor*) and *SCO3933* (*pra*) have a common intergenic region in which two binding sites were identified (Figure 1). Regions protected in footprint experiments are large (28 and 49 bp). Site K2 covers the transcriptional start site (TSS) of *SCO3933* identified by Jeong et al. 2016 [25], which is consistent with its probable role as a repressor. TSS of *SCO3932* is not known (detected was only an internal transcript), however, the distance of 38 bp between the start codon of *SCO3932* and the binding site K1 makes it likely that the protein binding influences the transcription of this gene. 

### 3.2. Participation of SCO3932 in Secondary Metabolism Regulation

In this work SCO3932 protein was found in *S. coelicolor* A3(2) cell lysate by binding to an intergenic DNA fragment containing the promoter sequence of the gene coding the first main polyketide synthase subunit CpkA and the promoter of co-transcribed tailoring genes. Park et al., 2009, in a similar experiment identified the same protein by binding to the promoter region of *actII-orf4* encoding the actinorhodin gene cluster activator, but the authors neither confirmed nor commented on that finding. 

We confirmed binding of recombinant His-tagged SCO3932 protein to the promoters of both *cpkD* and *actII-orf4* genes. We also found two binding sites in the intergenic region between *SCO3932* and *SCO3933* genes. SCO3932 binds to the promoters with different affinity. The strongest bindings occur to p3932/3933 fragments. Final information about DNA binding by the native SCO3932 protein in vivo comes from the ChIP-PCR experiment with anti-SCO3932 polyclonal antibodies. All three fragments were present in the DNA recovered from the sample from the *SCO3932* overexpression strain. In the sample from the M145 strain, only the *SCO3932/SCO3933* promoter fragment was clearly detected by PCR. This result is consistent with the strongest binding of p3932/3933 fragments observed in vitro.

SCO3932 was captured as a potential regulator of CPK (this work) and ACT [23] biosynthetic gene clusters, therefore we were interested in its effect on the respective compounds production and target promoter activity. Since gene deletion was unsuccessful, suggesting the potential role of SCO3932 as a Kor element in the kil-kor system, the work was focused on the effects of the protein overproduction. Activity of both tested promoters, pcpkD and pactII-orf4, was enhanced in the overexpression strain. Phenotypic observation showed enhanced ACT and unchanged CPK production (Figure 6b). In the case of ACT, increased production of this metabolite is consistent with enhanced production of the cluster direct activator ActII-orf4, as indicated by its promoter activity. CpkD is not a regulator, but one of post-polyketide tailoring enzymes [8], therefore its increased production may not change the overall CPK production. 

The SCO3932 protein belongs to a broad family of GntR-like regulators which have both DNA-binding and ligand-binding domains. Binding of an effector molecule changes the affinity of the protein to DNA and influences the regulatory effect. An interesting example is FadR from *Escherichia coli*, which, when bound to promoter regions, acts both as a repressor and activator of fatty acid degradation and biosynthesis genes, respectively. Binding of a long chain acyl-CoA by FadR leads to its dissociation from DNA and to de-repression of fatty acid degradation and inhibition of biosynthesis [31]. On the contrary, binding citrate molecules by CitO from *Enterococcus faecalis* strengthens its DNA binding which leads to activation of the citrate utilization pathway [32]. HutC subfamily members are involved in a variety of processes from amino acid uptake to plasmid transfer. UTRA domain ligand-binding pockets in these transcription factors have been adapted to accommodate diverse small molecules [21]. It is likely that the effects of SCO3932 overproduction observed here depend on the presence/absence of its specific ligand, however it is not possible to theoretically predict its identity. The effector molecule of SCO3932 homologue, KorSA, has not been identified so far. 

To our knowledge, this is the first report in which a regulator from an AICE is involved in the secondary metabolism regulation. There is an open question: whether this connection is important in the process of AICE excision/integration, and possibly controlled by the kil-kor system with SCO3932 as a potential Kor element.

## 4. Materials and Methods

### 4.1. DNA Manipulation and Bacterial Strains Growth Conditions

DNA manipulations were carried out according to standard protocols by Sambrook and Russel, 2001 [33]. All the PCR amplified fragments were first cloned into either p-GEM-T Easy or pTZ57R/T vectors to facilitate their digestion with restriction enzymes and further cloning into appropriate plasmids. All the PCR-derived clones were verified by DNA sequencing. Primers used are listed in Table 2. Oligonucleotides were obtained from Genomed and enzymes were supplied by Thermo Fisher Scientific (Waltham, MA, USA) Bacterial strains and plasmids used in this study are listed in Table 3. Culture conditions, media, antibiotic concentrations, transformation and conjugation methods followed the general procedures, for *E. coli* according to Sambrook and Russel, 2001 [33], and for *Streptomyces* as in Kieser et al., 2000 [34]. For phenotypic observations approx. 1.4 × 107 spores were streaked out in a square 1.5 × 1.5 cm on the medium 79 and 79NG (79 without glucose) [6]. If required, the media were supplemented with thiostrepton in the final concentration 2, 4 and 8 μg/mL. For CPK and ACT observation, plates were photographed after 26 h and 72 h, respectively.

### 4.2. DNA Affinity Chromatography

Streptomyces mycelia from liquid cultures were suspended in PBS with the addition of Protease Inhibitor Cocktail (Sigma-Aldrich, St. Louis, MO, USA) and lysed with UP200S ultrasonic disintegrator (Hielscher, Teltow, Germany) and centrifuged (16,000 rcf, 15 min, 4 °C). Biotinylated DNA fragment (30 pmoles) was bound to 2 mg of streptavidin coated magnetic beads M-PVA SAV2 (PerkinElmer Waltham, MA, US) and incubated for 1.5 h with 30 mg of total protein from cell lysate in 10 mL volume of 0.33 × PBS on a roller in room temperature. Magnetic beads were separated on the magnetic stand and washed four times with the same buffer. Proteins were eluted with 200 μL of 150 mM NaCl, precipitated with TCA-DOC and analyzed by SDS-PAGE. A gel slice containing an additional band, not visible in the control, was sent for identification by mass spectrometry. MS analysis was completed by the Mass Spectrometry Laboratory of the Institute of Biochemistry and Biophysics, Polish Academy of Sciences (Warsaw, Poland).

### 4.3. SCO3932 Protein Overproduction in E. coli

The *SCO3932* gene was PCR amplified using the primer pair Sc3932FW, Sc3932RV and cloned into NdeI and HindIII sites of pET21b (+) vector to give the plasmid pMZ16. *E. coli* BL21(DE3)pLysS strain containing pMZ16 was grown in LB medium with ampicillin (100 µg/mL) and chloramphenicol (34 µg/mL) to an OD600 of 0.5 at 37 °C. Expression was induced with 0.1 mM isopropyl-b-D-thiogalactopyranoside (IPTG) at 37 °C for 3 h. After harvesting by centrifugation, the cells from 2 L culture were lysed in 35 mL of binding buffer (300 mM NaCl, 50 mM Na3PO4 pH 8.0) by sonication and centrifuged at 4 °C, 26,890 rcf, 30 min. The clarified lysate was mixed with 1.5 mL of His-Select HF Nickel Affinity Gel (Sigma-Aldrich, St. Louis, MO, USA), the volume was adjusted to 50 mL with the buffer. Protein binding was performed at room temperature with gentle agitation for 30 min. The resin was collected by centrifugation (4 °C, 500× *g*, 5 min) washed with the binding buffer containing 10 mM imidazole and transferred to a column. His-tagged SCO3932 protein was eluted in the binding buffer containing 250 mM imidazole. The eluted fractions were examined using SDS-PAGE. Glycerol was added to 10% to the fractions and the protein was stored at −70 °C.

### 4.4. Electrophoretic Mobility Shift Assay (EMSA)

Promoter probes were labelled with IRDye 800 fluorescent compound. Amplified promoter fragments were cloned directly into the pTZ57R/T vector and used as templates for amplification with the labelled primers pTZBAM-800 and pTZXBA-800 complementary to the sequences flanking the cloning site of pTZ57R/T plasmid. Templates to amplify fragments for EMSA are listed in Table 3. Each sample contained 0.03 pmoles of labeled DNA in 16 μL binding buffer (10 mM Tris-HCl pH 7.5, 50 mM KCl, 1 mM DTT) including 1 μL of 100 ng/μL herring sperm DNA, and 1 μL of DTT/Tween solution (25 mM DTT, 2.5% Tween20) and different amounts of His-tagged SCO3932 protein. Samples were incubated for 20 min at room temperature. 4 μL of 40% sacharose was added and samples were loaded on nondenaturing 4% polyacrylamide–Tris–borate–EDTA gels. The results were visualized on LI-COR Odyssey apparatus with Image Studio Lite software. 

### 4.5. DNaseI Footprint

For the footprinting experiment the promoter fragments were uniquely radiolabeled at one end during amplification. One of the primers was labeled on its 5′ end with [32P]-ATP using T4 polynucleotide kinase. A binding mixture contained the radiolabeled fragment (approx. 200 cps) in 20 μL binding buffer (10 mM Tris-HCl pH 7.5, 50 mM KCl, 1 mM DTT) and a variable amount of His-tagged SCO3932 protein. The samples were incubated for 20 min at 25 °C and 10 min at 30 °C. For digestion, 2.5 μL of 10 times reaction buffer with MgCl2 for DNase I (Thermo Fisher Scientific, Waltham, MA, USA) and 2.5 μL of DNase I diluted in 1 x reaction buffer (0.005 u/μL) were added, and samples were incubated for exactly 5 min at 30 °C. The reaction was stopped by addition of 25 μL of Stop buffer (200 mM NaCl, 100 mM EDTA, 1% SDS) and incubation for 10 min at 75 °C. DNA from each sample was extracted with phenol/chloroform/isoamyl alcohol (25:24:1) and ethanol-precipitated. The precipitated DNA was dissolved in loading buffer (95% formamide, 20 mM EDTA, 0.05% bromophenol blue, 0.05% xylene cyanol FF) and loaded onto a denaturing 8% polyacrylamide–Tris–borate–EDTA gel. Dideoxy sequencing ladders were generated using Thermo Sequenase Cycle Sequencing Kit (Affymetrix, St.Clara, CA, USA) and the same labeled primers as those used to prepare the probes.

### 4.6. DMS Footprint

In vitro DNA fragments were modified as described previously by Sasse-Dwight [39]. DMS-methylated DNA’s were used as templates in primer extension (PE) reactions containing the same primers as in DNAse I experiment. The gels were scanned using a Typhoon FLA 9500 (GE Healthcare, Chicago, IL, USA) and analyzed using ImageQuant software.

### 4.7. Chromatin Immunoprecipitation (ChIP)

To obtain anti-SCO3932 polyclonal antibodies, rabbits were immunized three times at three-week intervals with intradermal injections of 0.5 mg of SCO3932 protein in PBS dispersed with 0.5 mL of Freund’s adjuvant. Blood serum was collected after an additional 9 weeks. IgG fraction was obtained from serum by salting-out with 20% ammonium sulphate followed by dialysis against 0.5 M Tris–HCl pH8.0. In ChIP procedure 24 μg of the IgG fraction of antibodies were added to each sample.

*Streptomyces* strains M145 and P171 were grown for 19 h in 35 mL of liquid 79NG medium inoculated with 10^8^ spores. In case of P171 thiostrepton was added at the concentration of 8 μg/mL. Chromatin immunoprecipitation was performed as previously described by Al-Bassam et al. [40] and followed by PCR testing of the presence of specific DNA fragments (primer pairs are given in Table 2). Total genomic DNA extracted from parallel cultures was used as a control.

### 4.8. Construction of Overexpression and Deletion Mutant Strains

Gene *SCO3932* was cloned into a multicopy plasmid pWP3 under the control of ermEp* [26] constitutive promoter and into pIJ6902 overexpression vector, under the thiostrepton inducible promoter PtipA, resulting in pKL20 and pKL21 plasmids, respectively (Table 3). The two constructs were introduced into *S. coelicolor* M145 by conjugation with *E. coli* ET12567/pUZ8002 containing appropriate vectors giving rise to P170 and P171 strains, respectively (Table 2). 

An attempt of *SCO3932* gene deletion was performed by homologous recombination with plasmid pMB14. The construct was prepared as follows: Upstream flanking arm was PCR amplified using the primer pair 3932LRFw and 3932LRFw and cloned into BamHI and EcoRI sites of pOJ260 to give pMB12 plasmid. Downstream flanking arm was amplified using the primer pair 3932PRFw and 3932PRRv and cloned into PstI and BamHI sites of pMB12 to give pMB13. Finally, a neomycin resistance cassette was cloned into the XbaI site of plasmid pMB13. Prepared plasmid was introduced into *S. coelicolor* M145 by conjugation with *E. coli* ET12567/pUZ8002 strain containing pMB14. Transconjugants were screened for neomycin resistance and apramycin sensitivity as an indication of a double crossover event.

### 4.9. Promoter Activity Measurement

For the promoter activity assay in luciferase reporter system, plasmids pFLUXH-pcpkD and pFLUXH-pactII-orf4 (Table 3) were introduced by conjugation into *S. coelicolor* P170 (SCO3932 overproducing) and P123 (control with empty plasmid) strains. Luciferase activity was measured in the ClarioStar Plus microplate reader (BMG Labtech) as described before [12]. Strains with pFLUXH-pcpkD and pFLUXH-pactII-orf4 plasmids were grown on solid medium 79NG and 79 for *cpkD* and *pactII-orf4* promoters, respectively. 

## Figures and Tables

**Figure 5 ijms-22-11867-f005:**
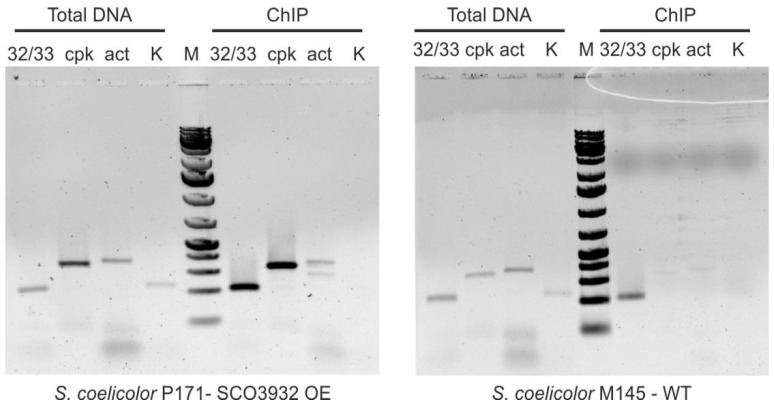
Chromatin immunoprecipitation. Nucleoprotein complexes crosslinked with glutaraldehyde were precipitated by anti-SCO3932 polyclonal antibodies, and DNA fragments were amplified by PCR with primers listed in Table 2. SCO3932 overexpression in P171 was induced by thiostrepton (8 μg/mL). 32/33—SCO3932/SCO3933 promoter, cpk—*cpkD* promoter, act—*actII-orf4* promoter, K—control fragment, M—1 kb Plus DNA Ladder (Thermo Fisher Scientific).

**Figure 6 ijms-22-11867-f006:**
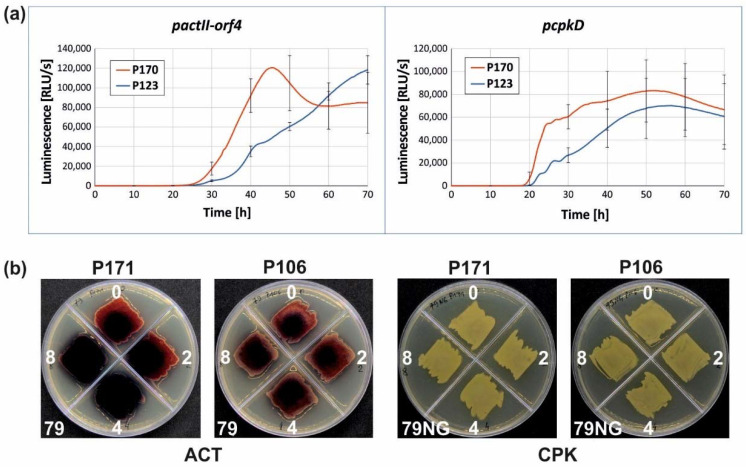
Effects of SCO3932 overexpression. (**a**) Activity of *cpkD* and *actII-orf4* promoters measured in the luciferase reporter assay, P170—strain overexpressing SCO3932, P123—control strain, standard deviation was shown for every 10 h time-point for clarity; (**b**) Phenotypic effects of SCO3932 overexpression induced by thiostrepton in the concentration 0, 2, 4, 8 μg/mL (as indicated) on 79 and 79NG media, P171—strain overexpressing SCO3932, P106—control strain.

**Table 1 ijms-22-11867-t001:** *S. coelicolor* A3(2) regulatory proteins interacting with bait fragments, detected by mass spectrometry.

Nr	Protein Name	Mass [Da]	Score	Matches	emPAI *
1	hypothetical protein SCO7102	19196	987	19	2.40
2	transcriptional regulator GntR type SCO3932	28559	386	13	2.15
3	DNA binding protein SCO1926	27119	112	2	0.25
4	transcriptional regulator IclR type SCO1872	28494	107	2	
5	DNA binding protein SCO6003	31958	94	1	

* Exponentially Modified Protein Abundance Index.

**Table 2 ijms-22-11867-t002:** Oligonucleotides used in this work. Gene numbers according to Bentley et al. 2002 [5]. Restriction sites are in bold.

Primer	Sequence	Amplified Fragment; Description
Sc3932FW	GGATCCCATATGACGGAGATCCAGCGC	gene *SCO3932;* for expression
Sc3932RV	GAATTCTCAAAGCTTGGTCTGGCGGCGTCGGGT	
3932LRFw	AAGCTTACGTGGTGAACCATGCAGT	*Upstream* flanking arm for deletion of *SCO3932*
3932LRRv	CTGCAGAGACCTGACCGGCGA CGA	
3932PRFw	GGATCCGGCGAACTCGCCCGAAAG	*Downstream* flanking arm for deletion of *SCO3932*
3932LRFw	AAGCTTACGTGGTGAACCATGCAGT	
p3932FW	GGGAATGGACTGCATTTCTG	p3932/3933; *SCO3932* and *SCO3933* intergenic region for EMSA
p3932RV	GAACTCGCCCGAAAGGAT	
FPrv3932	AAGAATTCTGGATCTCCGTCATCTC	detection of *SCO3932* and *SCO3933* intergenic region in ChIP-PCR experiment, primers used for footprint
FPfw3932	AAGGATCCGAATGGACTGCATTTCTG	
CTR1-BT	biotin-TCGCCTGGAGAACGGGCC	control fragment, biotinylated
CTR2	CGGAATTCGTCGACGGAGCCACCGGCTTC	
CTR3	CGTTAAATGCCTGGACTGTG	control fragment in ChIP-PCR experiment
CTR4	TGTAGCGGATGCCGATGTT	
AD1-BT	biotin-GTGACGTTCGCGAAGGTCTCG	pcpkA/D-BT; *cpkA* and *cpkD* intergenic fragment, biotinylated
AD2	GACAGTCCCACGACAGCGATC	
AD3	CGACCCGAATCCTCTTCCAGA	primer for footprint
AD4	GAAGGTCTCGGAGAGAGCAC	pcpkA/D; *cpkA* and *cpkD* intergenic fragment for EMSA, AD4 primer was used for footprint
AD5	CAGGACAGTCCCACGACAG	
AD6	GGAGAAAAGGCCCGCCATGCT	primer used together with AD3 for detection of *cpkA* and *cpkD* intergenic region in ChIP-PCR experiment,
pact-fw	CGCTCGCCCGGCGCGAGGACCCTTC	pactII-orf4; promoter of gene *actII-orf4* for EMSA, primers used for footprint and for ChIP-PCR experiment
pact-rv	TGAGGAGCAGCAGCACCAGGAGCTG	
pactII-orf4_Bam_F	GGATCCCTGCTGATCGCGAGCGTGG	promoter of gene *actII-orf4* for cloning into reporter vector pFLUXH
pactII-orf4_Nde_R	CATATGCGCCCCCGTCGAGATTCTC	
pTZBAM-800	IRD800-ATGCAGGCCTCTGCA	IRDye 800 labeled primers flanking the cloning site of pTZ57R/T vector
pTZXBA-800	IRD800-TCGGTACCTCGCGAA	

**Table 3 ijms-22-11867-t003:** Bacterial strains and plasmids used in this study.

Strain or Plasmid	Relevant Genotype or Description	Source or Reference
** *E. coli* **		
DH5α	F^-^ *endA1 glnV44 thi-1 recA1 relA1 gyrA96 deoR nupG* Φ80d*lacZ*ΔM15 Δ(*lacZYA-argF*)U169, *hsdR17*(r_K_^-^ m_K_^+^), λ^–^	Promega
BL21(DE3)pLysS	F-, *ompT*, *hsdSB* (rB-, mB-), *gal*, *dcm* (DE3), pLysS (Cam^R^)	Promega
ET12567/pUZ8002	strain for conjugal transfer of DNA from *E. coli* to *Streptomyces* (*dam dcm hsdS* Cam^R^ Tet^R^ on the bacterial chromosome; *tra* Kan^R^ RP4 23 on pUZ8002)	[34]
***S. coelicolor* A3(2)**		
M145	wild type strain, a plasmidless variant of *S. coelicolor* A3(2) (SCP1- SCP2-)	[34]
P170	overproduction of SCO3932 from a strong constitutive promoter *ermEp** on a multicopy plasmid (M145 + pKL20)	this work
P123	control with empty plasmid (M145 + pCJW93)	this work
P171	overproduction of SCO3932 from a thiostrepton inducible promoter P*tipA* on an integrative plasmid (M145 + pKL21)	this work
P106	control with empty plasmid (M145 + pIJ6902)	this work
M145 and P171 derivatives for luciferase reporter assay	M145 and P171 harbouring either pFLUXH-pcpkD or pFLUXH-pactII-orf4	this work
**Plasmids**		
pTZ57R/T	T-vector from InstT/A Cloning kit for direct cloning of PCR products	Thermo Fisher Scientific
pGEM-T Easy	T-vector for direct cloning of PCR products	Promega
pET21b	plasmid for expression of proteins with C-terminal His-tag (T7 promoter, Amp^R^)	Novagen
pCJW93	high copy number plasmid, Apra^R^ (*aac3(IV)*), *oriT* (RK2), ThioR (*tsr*), P*tipA*,	[35]
pWP3	pCJW93 in which P*tipA* promoter was cut out with NdeI i DraI and replaced with *ermEp** promoter from pIJ10257 plasmid (digested with KpnI, 3′ overhang blunted, digested with NdeI), Apra^R^ (*aac3(IV)*), *oriT* (RK2), Thio^R^ (*tsr*), promotor *ermE*p*)	this work
pIJ6902	integrative vector, Apr^R^ (*acc3(IV)*), *oriT* (RK2), Thio^R^ (*tsr*), P*tipA*	[15]
pIJ10257	ΦBT1 integrating overexpression plasmid containing strong constitutive promoter *ermEp**, Hyg^R^	[36]
pMZ10	gene *SCO3932* amplified with primers Sc3932FW and Sc3932FW in pGEM-T Easy	this work
pMZ16	gene *SCO3932* cut out from pMZ10 with NdeI and HindIII cloned in the corresponding sites of pET21b	this work
pKL20	gene *SCO3932* cut out from pMZ10 with NdeI and EcoRI cloned in the corresponding sites of pWP3	this work
pKL21	gene *SCO3932* cut out from pMZ10 with NdeI and EcoRI cloned in the corresponding sites of pIJ6902	this work
pOJ260	vector for conjugal transfer from *E. coli* to *Streptomyces*, non-integrative, (Apra^R^), does not propagate in *Streptomyces*	[37]
pMB12	pOJ260 with the *Upstream* flanking arm	this work
pMB13	pOJ260 with the *Upstream* and *Downstream* flanking arms	this work
pMB14	pOJ260 with the neomycin resistance cassette cloned between the *Upstream* and *Downstream* flanking arms to replace *SCO3932* gene	this work
pTZ-pactII-orf4	promoter of *actII-orf4*, amplified with primers pact-fw and pact-rv, in pTZ57R/T, used as template to amplify IRD labeled pactII-orf4 fragment for EMSA	this work
pTZ-pcpkA/D	fragment pcpkA/D, amplified with primers AD4 and AD5, in pTZ57R/T	this work
pTZ-p3932	fragment p3932/33, amplified with primers FPrv3932 and FPfw3932, in pTZ57R/T	this work
pTZ57R-T-pactII-orf4	promoter of *actII-orf4*, amplified with primers pactII-orf4_Bam_F and pactII-orf4_Nde_R, in pTZ57R/T	this work
pFLUXH	ΦBT1 integrating reporter plasmid with a promoterless luciferase operon *luxCDAEB*	[38]
pFLUXH-pcpkD	pFLUXH containing promoter region of *cpkD*	[12]
pFLUXH-pactII-orf4	promoter region of *actII-orf4* cut out from pTZ57R-T-pactII-orf4 with BamHI and NdeI and cloned into pFLUXH	this work

## Data Availability

Not applicable.

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
