# Peer review of "GntR-like SCO3932 Protein Provides a Link between Actinomycete Integrative and Conjugative Elements and Secondary Metabolism"

_ijms, 2021, doi:10.3390/ijms222111867_

Round 1

Reviewer 1 Report

In this work, the authors have studied the role of the GntR like SCO3932 protein using molecular tools. This is a good work and I recommend it for publication, but the authors should correct how the bacterial strains are written (the bacterial strains must be written with italic letters) 

Author Response

The authors are grateful to this reviewer for reading of the manuscript, positively evaluating it, and suggesting improvements. Suggested changes have been incorporated into a revised version of the manuscript.

Reviewer 2 Report

In the paper of Pawlik et al. entitled « GntR like SCO3932 protein provides a link between actinomycete integrative and conjugative elements and secondary metabolism. » the authors convincingly demonstrated, by different means, that the KorSA-like regulator SCO3932, present in a region of S. coelicolor related to the integrative and replicative plasmid pSAM2 from S. ambofaciens, is able to interact strongly, as expected, with the promoter region of its divergent gene sco3933 encoding a Par-like protein but also, but more weakly, with the promoter regions of actII-orf4, the activator of the ACT cluster as well as with the DNA region located between cpkA encoding the major synthase sub-unit of the coelimycin biosynthetic cluster CPK and the divergent operon cpkDEG participating in tailoring re-actions. Furthermore, the authors demonstrated that in condition of over-expression sco3932 has a positive impact on the transcription of actII-orf4 and of cpkD and this was correlated with an increase of the production of ACT but not of that of CPK.

This study is the first one to reveal that a regulator of plasmid functions has also potentially a positive impact of the biosynthesis of antibiotics such as that of the blue polyketide Actinorhoddin/ACT. The biological signification of the link between these two functions is still unclear. The onset of ACT biosynthesis was shown to coincide with an abrupt drop in the intracellular ATP concentration (https://www.nature.com/articles/s41598-017-00259-9) and this was attributed to the ability of ACT to capture electrons of the respiratory chain with its quinone groups (“anti-respiratory function” of ACT that is related to its “anti-oxydant” function, https://www.nature.com/articles/s41598-020-65087-w and  https://www.mdpi.com/2079-6382/9/2/83). Since plasmid transfer gives rise to a zone of growth inhibition of the recipient bacteria called “pocks”, it is possible that ACT, that has a negative impact on ATP generation, contributes to this growth arrest.

Author Response

The authors thank the reviewer and are sincerely grateful for the in-depth reading of the manuscript. As suggested, the manuscript was proofread by a native speaker. Changes were made to the revised version of the manuscript.